# Optimizing assessment of low frequency H-reflex depression in persons with spinal cord injury

Charles J. Creech [1,2], Jasmine M. Hope [3], Anastasia Zarkou[1], Edelle C. Field-Fote[1,2,3]*

1 Crawford Research Institute, Shepherd Center, Atlanta, Georgia, United States of America, 2 Program in Applied Physiology, Georgia Institute of Technology, School of Biological Sciences, Atlanta, Georgia, United States of America, 3 Division of Physical Therapy, Department of Rehabilitation Medicine, Emory University School of Medicine, Atlanta, Georgia, United States of America

* Edelle.Field-Fote@shepherd.org

**Data Availability Statement:** All relevant data are within the paper and its Supporting information files.

**Funding:** This study was funded by the NIH National Institute of Child Health and Human

## Abstract

Considering the growing interest in clinical applications of neuromodulation, assessing effects of various modulatory approaches is increasingly important. Monosynaptic spinal reflexes undergo depression following repeated activation, offering a means to quantify neuromodulatory influences. Following spinal cord injury (SCI), changes in reflex modulation are associated with spasticity and impaired motor control. To assess disrupted reflex modulation, low-frequency depression (LFD) of Hoffman (H)-reflex excitability is examined, wherein the amplitudes of conditioned reflexes are compared to an unconditioned control reflex. Alternatively, some studies utilize paired-pulse depression (PPD) in place of the extended LFD train. While both protocols induce similar amounts of H-reflex depression in neurologically intact individuals, this may not be the case for persons with neuropathology. We compared the H-reflex depression elicited by PPD and by trains of 3–10 pulses to an 11-pulse LFD protocol in persons with incomplete SCI. The amount of depression produced by PPD was less than an 11-pulse train (mean difference = 0.137). When compared to the 11-pulse train, the 5-pulse train had a Pearson's correlation coefficient ($R$) of 0.905 and a coefficient of determination ($R^2$) of 0.818. Therefore, a 5-pulse train for assessing LFD elicits modulation similar to the 11-pulse train and thus we recommend its use in lieu of longer trains.

## Introduction

Following spinal cord injury (SCI), impaired modulation of spinal reflex excitability is associated with spasticity [1]. Increasingly, electroceutical interventions, such as transcutaneous and epidural spinal stimulation, are being used to counter this impaired modulation [2] and reduce spasticity [3–5]. Modulation of spinal circuits through these interventions has been associated with improved functional outcomes and mobility [6, 7]. As use of these electroceutical and other neuromodulatory approaches grows, it is important to optimize tools to assess their efficacy.

Development (NICHD; https://www.nichd.nih.gov) R01HD079009 (E.C.F.-F.). Financial support was provided by the Jack and Dana McCallum Neurorehabilitation Training Fellowship (C.J.C.) during manuscript development. The funders had no role in study design, data collection and analysis, decision to publish, or preparation of the manuscript.

**Competing interests:** The authors have declared that no competing interests exist.

The Hoffman (H)- reflex is a commonly used probe of spinal reflex excitability of monosynaptic spinal circuits [8], which is modulated in a rate-dependent manner via post activation depression (PAD) [9]. PAD of the H-reflex is often assessed via low frequency depression (LFD) using repeated pulses of stimulation [10–14]. LFD attenuates H-reflex amplitude in people who are neurologically intact [15], however, in persons with SCI, the magnitude of H-reflex modulation resulting from LFD is reduced [16]. The result of this reduced reflex modulation is enhanced synaptic excitation, which is commonly manifested as spasticity.

The LFD protocol consists of a train of pulses delivered at a specific interpulse interval, with each pulse eliciting a reflex response. The response to the first pulse is considered the control response to which the subsequent conditioned responses are compared. LFD protocols typically require an extended testing period of 10 [16, 17] to 20 pulses per train [18]. If multiple trains are completed, this may result in hundreds of stimuli being applied to a participant. This extended testing period imposes a burden on participants, many of whom must be mindful of their skin health. Furthermore, LFD protocols can elicit clonus or other reflex responses in persons with SCI, compromising data quality [19].

Paired-pulse depression (PPD) is an alternative approach to assessing H-reflex modulation in which only 2 H-reflexes are elicited and their amplitudes compared. A study in neurologically intact individuals demonstrates that PPD yields comparable depression to a 10-pulse LFD train [17]. Such a protocol overcomes the disadvantages of the LFD protocol and may be a valuable tool for assessing neuromodulation. However, if the PPD protocol is to be used as an alternative to the LFD protocol it is first necessary to determine whether the two protocols yield similar outcomes in individuals with neurologic conditions in whom reflex modulation is impaired.

The purpose of this study was to compare, in participants with incomplete SCI, the magnitude of reflex depression elicited by a typical 11-pulse LFD protocol, a typical PPD protocol, and stimulation trains of varying lengths. Identifying the minimum number of pulses required to obtain a reliable estimate of LFD could reduce the time requirement and discomfort associated with assessing efficacy of neuromodulation interventions in participants with neurologic conditions.

## Materials and methods

This study was conducted with ethical approval from the Shepherd Center Research Review Committee. All participants gave written informed consent prior to study enrollment, which was conducted in accordance with the guidelines of the Declaration of Helsinki. The current study was a supplemental analysis of data collected as part of the second phase of a larger study focused on physical therapeutic spasticity interventions for persons with SCI that was registered with clinicaltrials.gov (NCT02340910). Participant recruitment for this phase began on March 20th, 2017 and concluded on March 5th, 2020.

### Participants

Individuals were eligible for participation in the larger study if they met the following inclusion criteria: 16–72 years of age, $\geq$ 6 months since time of SCI, neurologic level of injury at or above spinal level T12, able to sit at the edge of a mat without the assistance of another person, have mild spasticity affecting muscles in at least one lower extremity (as determined by participant self-report), and are able to provide own consent. The current study analyzed data from participants who met the additional inclusion criterion of having a measurable H-reflex that met our amplitude criteria during electrophysiological testing. Individuals were excluded for the following reasons: progressive or potentially progressive spinal lesions, history of

cardiovascular irregularities, orthopedic conditions that would limit their participation in the protocol (e.g., knee or hip flexion contractures > 10°), or problems following instructions.

## Electrophysiologic measurements

Electrophysiologic tests were performed with participants in a comfortable, semi-reclined position on an adjustable height mat with both legs fully extended. Participants were instructed to remain as relaxed and still as possible throughout the testing. The more spastic leg as self-reported by participants was assessed during testing. Participants' skin was cleansed with alcohol and treated with an abrasive paste (Nuprep, Weaver and Company, Aurora, CO, USA) to improve adhesion and to reduce skin resistance prior to placement of pre-amplified (differential) electromyography (EMG) sensors (MA411, Motion Lab Systems, Baton Rouge, LA; Input Impedance > 100,000,000 ohms, CMR > 100 dB at 65 Hz; pre-amplification x20) and stimulating electrodes. The EMG sensors had a body size of 38 x 19 x 8 mm and consisted of two stainless steel 12-millimeter (mm) disks separated by a 13 x 3 mm bar. The inter-electrode distance was 17 mm. EMG sensors were placed over the distal portion of the tibialis anterior muscle belly and over the soleus muscle at the midline of the posterior calf between the heads of the gastrocnemius. EMG sensor placement was verified and activity was recorded (sampling rate of 1,000 Hz) using commercial software (Spike2 version 8.02e; Cambridge Electronic Design Limited, Cambridge, England) at a gain of 1.

Fabric stimulating electrodes were placed over the tibial nerve in the popliteal fossa (cathode; 1.25" diameter) and over the center of the patella (anode; 2.0" diameter). Monophasic isolated square-wave pulses with a pulse width of 1 millisecond (ms) produced by an electrical stimulator (Grass S88x, Grass Technologies, West Warwick, RI) were used to trigger a constant-current stimulator (Digitimer DS7A, Digitimer, Hertfordshire, UK). The initial stimulus began with an intensity of 5 milliamperes (mA), increasing by 2–3 mA until an H-reflex was elicited. To avoid rate-sensitive effects of stimulation, the inter-pulse interval was 10 seconds. The position of the stimulating electrode was adjusted to obtain the largest peak-to-peak (µV) H-reflex at a minimal intensity. Reflexes were observed using oscilloscopic software (Signal version 6.02; Cambridge Electronic Design Limited, Cambridge, England) and were not filtered. An H-reflex recruitment curve was then obtained by increasing stimulus intensity by 2.5–5.0 mA until a plateau of the M-wave (M-max) was identified. The stimulus intensity corresponding to 10–30% of the M-max amplitude was identified and was utilized for the LFD trains. This range was selected as it represents the ascending portion of the H-reflex recruitment curve where the threshold difference between various motor units is smallest [20], allowing for efficient facilitation or inhibition of the reflex [8].

The data collected during this electrophysiologic testing was utilized for the calculation of LFD, PPD, and depression of abbreviated trains. The H-reflex evoked by the first stimulus was considered the control reflex (H1). The 10 reflexes elicited following H1 were considered conditioned reflexes (H2, H3, etc.). Each LFD train consisted of 11 stimuli delivered with a 1- second inter-pulse interval (i.e., 1 Hz). While previous studies have demonstrated that higher frequencies may result in greater reflex depression [20], they can also be associated with increased muscle spasms in people with SCI [19]. The M-waves accompanying reflex responses were also recorded. In total, each participant underwent at least 10 LFD trains, with some completing more to account for trains in which muscle spasms occurred or if H1 did not fall within 10–30% of the M-max amplitude. To accommodate variability in control H-reflex amplitudes, the first 4 trial sequences with H1 amplitudes between 10% and 30% of the M-max were analyzed.

To determine the magnitude of depression associated with the 11-pulse LFD train, the conditioned reflex responses evoked by pulses H2 –H11 were averaged and normalized to H1 (average of H2-11/H1). To determine the magnitude of depression associated with PPD, the conditioned reflex response elicited by H2 was normalized to H1 (H2/H1). Likewise, this procedure was used to determine the magnitude of depression associated with abbreviated trains of different lengths (i.e., average of H2-3/H1, average of H2-4/H1. . . average of H2-10/H1).

### Data analysis

All data acquired were continuous. Statistical tests were conducted using commercially available software (SPSS Version 28, IBM Corporation, Armonk, NY USA). Confidence intervals (95%) were calculated using Excel (Version 2312, Microsoft Corporation, Redmond, WA USA). The mean and standard deviation are displayed as (mean ± standard deviation).

**M-wave reliability.**   Based on the sample size utilized during the analysis of the M-waves, non-parametric tests were considered. To verify that the stimuli were recruiting similar numbers of motoneurons over the course of the conditioning train, a Friedman test was performed to assess the effect of number of elicited reflexes on M-wave amplitude.

**Optimal pulse train for LFD estimation in SCI.**   Pearson's correlation coefficient ($R$) was used to assess the strength of the relationship between the conditioned responses elicited by 11-pulse LFD train and the conditioned responses elicited by PPD and by the abbreviated trains. Correlation coefficients were classified as negligible (0.00 to 0.30), low (0.30 to 0.50), moderate (0.50 to 0.70), high (0.70 to 0.90), or very high (0.90 to 1.00) [21]. In addition, mean difference and the proportions of variance ($R^2$) were calculated.

## Results

### Demographics

Of the 38 individuals enrolled in the larger study, 20 participants (Mean age = 50.25 ± 14.26 years; 10 males, 10 females) met the original inclusion criteria and had a reliable initial H-reflex that met the criteria of being between 10% and 30% of the maximum M-wave. Protocol deviations were completed for participant 71 (<6 months since injury) and 73 (> 72 years of age). Participant demographics for the participants included in the present study are given in Table 1. An H-reflex representative of those elicited during this study is presented in Fig 1.

**M-wave reliability.**   An M-wave representative of those elicited during this study is included in Fig 1. There were no significant differences between the M-wave amplitudes at each of the 11 pulses of the LFD train ($\chi^2(10) = 5.642$, p = 0.844), indicating that the effective stimulus intensity remained stable.

**Optimal pulse train for LFD estimation in participants with SCI.**   The correlation between the amount of depression elicited by the 11-pulse LFD train and by the abbreviated trains increased as the number of included pulses increased (Fig 2, Table 2). Increases in correlation corresponded with decreases in the mean difference between the amount of depression elicited by LFD and that elicited by the abbreviated trains. All abbreviated trains were significantly correlated with the 11-pulse LFD train ($p \leq 0.002$). Relative to the depression elicited by the 11-pulse train, the shortest train eliciting depression with a *high* correlation was the 3-pulse train (H1 and 2 conditioning pulses; R(18) = 0.752, p < 0.001, $R^2$ = 0.566, mean difference = 0.087). The shortest train eliciting depression with a *very high* correlation was the 5-pulse train (H1 and 4 conditioning pulses; R(18) = 0.905, p < 0.001, $R^2$ = 0.818 mean difference = 0.037). The 2-pulse train (PPD) produced a response amplitude that had the greatest difference when compared to LFD (mean difference = 0.137). The values for all other abbreviated trains in relation to the 11-pulse train are outlined in Table 2.

**Table 1. Demographics for all participants enrolled in the larger study.**

| Participant ID | Sex | Age (y) | Time since injury; y (m) | AIS | Neurologic Level of Injury |
|---|---|---|---|---|---|
| 38 | M | 54 | 6 (9) | D | C4 |
| 39 | F | 63 | 6 (7) | C | C7 |
| 40 | F | 21 | 1 (0) | D | T11 |
| 43 | M | 49 | 29 (5) | D | C5 |
| 46 | M | 59 | 5 (6) | D | C5 |
| 51 | F | 60 | 7 (2) | D | C5 |
| 52 | M | 61 | 2 (0) | D | C1 |
| 55 | M | 44 | 19 (11) | D | C2 |
| 59 | F | 39 | 1 (6) | C | C7 |
| 60 | F | 29 | 0 (11) | D | T8 |
| 61 | F | 54 | 1 (5) | C | C8 |
| 62 | M | 40 | 0 (8) | D | C4 |
| 65 | M | 53 | 0 (6) | D | C3 |
| 67 | M | 60 | 3 (3) | D | C5 |
| 68 | M | 44 | 8 (7) | D | T8 |
| 70 | F | 59 | 4 (8) | D | C7 |
| 71 | F | 58 | 0 (5) | B | T5 |
| 72 | M | 22 | 0 (6) | C | C4 |
| 73 | F | 75 | 6 (11) | D | C6 |
| 74 | F | 61 | 15 (3) | D | C5 |

Abbreviations: y, years; m, months; AIS, American Spinal Injury Association Impairment Scale

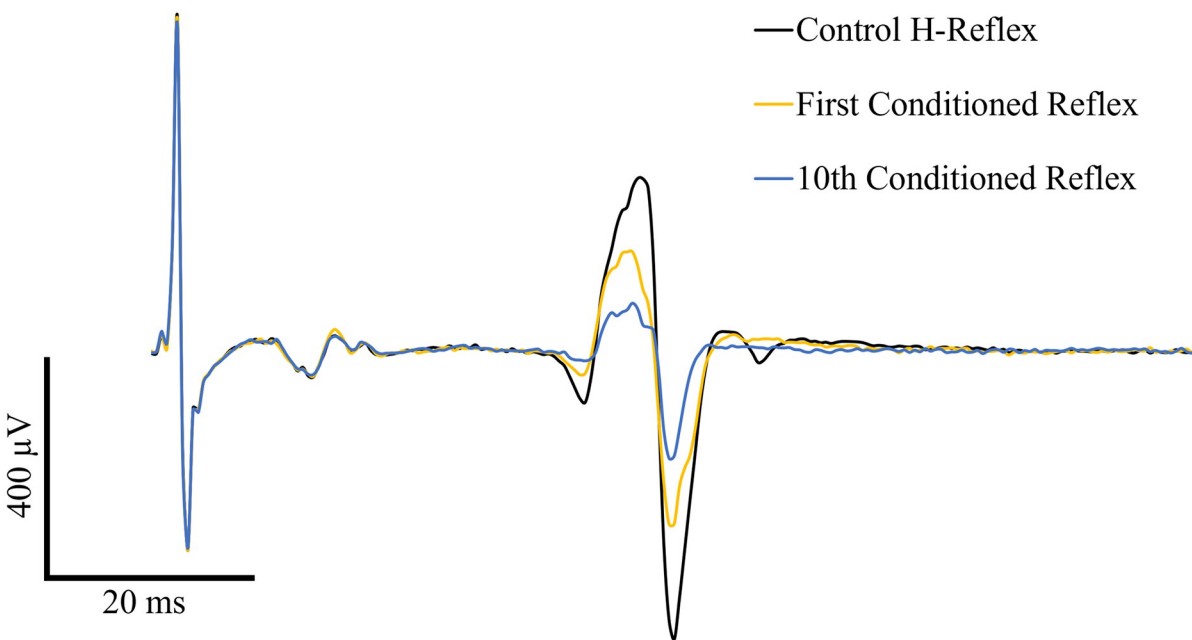

**Fig 1. H- and M- wave tracings from participant 59.** Tracings from the control H-reflex (H1), the first conditioned reflex (H2), and the 10th conditioned reflex (H11) are included.

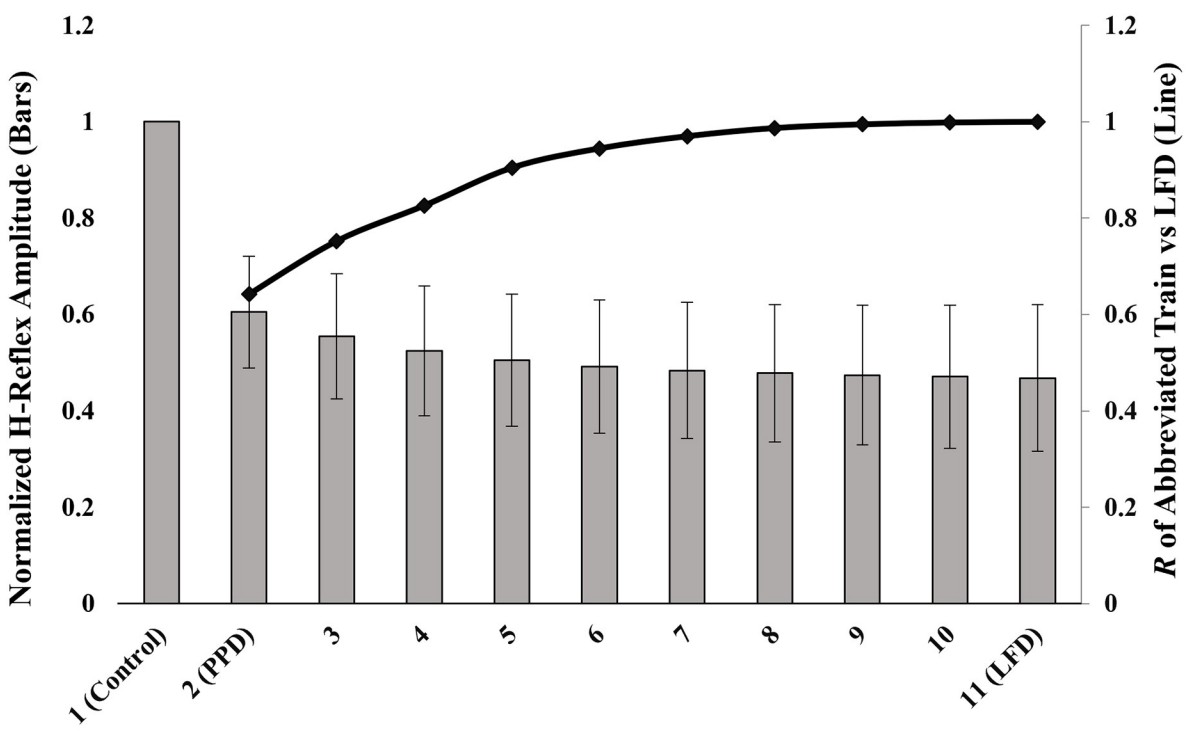

**Fig 2. Normalized H-reflex amplitudes for various abbreviated train lengths.** The displayed value on the primary (left) axis is an average of all denoted conditioned responses normalized to H1. Error bars represent standard deviation. The secondary (right) axis corresponds with the Pearson's correlation value between the abbreviated train and the 11-pulse LFD train.

**Table 2. Statistical values.**

| Abbreviated Train Length (number of stimuli) | Mean Difference vs LFD | 95% CI of Mean Difference vs LFD (LB, UB) | Pearson Correlation Coefficient (R) |
|---|---|---|---|
| 1 (Control) | 0.532 | 0.228, 0.836 | x |
| 2 (PPD) | 0.137 | -0.099, 0.373 | 0.642 |
| 3 | 0.087 | -0.116, 0.289 | 0.752 |
| 4 | 0.057 | -0.116, 0.229 | 0.826 |
| 5 | 0.037 | -0.092, 0.167 | 0.905 |
| 6 | 0.024 | -0.076, 0.124 | 0.945 |
| 7 | 0.016 | -0.059, 0.090 | 0.970 |
| 8 | 0.010 | -0.041, 0.061 | 0.987 |
| 9 | 0.006 | -0.027, 0.039 | 0.995 |
| 10 | 0.003 | -0.014, 0.020 | 0.999 |

Values associated with the comparison of the reflex amplitude occurring with each abbreviated train (as a proportion of the control reflex) compared to LFD via the 11-pulse train. Mean differences calculated as the average normalized amplitude of conditioned H-reflexes subtracted from that of LFD. Abbreviations: CI, confidence interval; LB, lower bound; UB, upper bound

## Discussion

LFD protocols are commonly utilized to assess spinal reflex excitability [16, 17], and are of great value for assessing changes in excitability associated with neuromodulation interventions. This study explored the relationship between the amount of H-reflex depression elicited by the typical 11-pulse LFD protocol, that elicited by PPD, and by pulse trains consisting of 3–10 pulses in persons with incomplete SCI. Depression elicited by all abbreviated trains was less than that elicited by LFD, however the depression elicited by all abbreviated trains were significantly correlated with that of LFD. As the number of included pulses increased, the mean difference between the amount of depression elicited by the abbreviated trains and the 11-pulse LFD train decreased (Table 2), with the greatest difference occurring with only one conditioned response.

Our findings are consistent with the prior work demonstrating that there is a cumulative decrease in the H-reflex with greater numbers of conditioning stimuli [22]. When compared to the 11-pulse LFD train, the depression elicited by the 5-pulse and 3-pulse trains were comparable, resulting in only slightly less depression (3.7% and 8.7%, respectively). The 5-pulse train was the shortest abbreviated train for which a very high correlation was identified, while the 3-pulse train had a high correlation. While Pearson correlation coefficients do not gauge whether sets of data are equivalent, they do offer a method of assessing the strength and direction of the relationships [23]. Based on these findings, the use of a 5-pulse LFD train offers insights about the amount of neuromodulation that is similar to that of the more common 11-pulse train.

Our findings are in contrast to prior research on neurologically intact participants, wherein the depression elicited by the PPD and LFD approaches with a 1-second interstimulus interval were found not to be statistically different, making PPD an appropriate substitute for multi-pulse LFD trains [17]. These results are likely attributable to impairment of reflex modulation in our participants, which is common in persons with SCI. Previous literature demonstrates that persons with SCI exhibit significantly less H-reflex depression than their neurologically intact counterparts [16]. Despite impairment of reflex modulation in persons with SCI, as with neurologically intact individuals, the greatest amount of depression occurred with the first conditioning pulse [17] (Table 2). The data from the current study suggests that, in persons with SCI, a larger number of pulses is required to elicit maximal reflex depression in comparison to individuals who are neurologically intact. In contrast, people who are neurologically intact demonstrate reflex depression that does not significantly change with repeated conditioning stimuli [17, 24].

As neuromodulation approaches to influence neural excitability become more broadly used, efficient and valid methods to assess their effects are increasingly important. The use of an abbreviated train of pulses to investigate neurologic modulability could alleviate some of the challenges inherent to electrophysiologic testing in persons with neurologic impairment. When collecting these data, participants are often required to sit/lie in one position remaining as still as possible. This can be challenging for people with SCI as muscle spasms or spasticity can cause involuntary movement [25] that can produce erroneous data or preclude their inclusion. Additionally, shorter pulse trains could cause less discomfort with stimulation, limiting the variations in descending drive that occur when nociceptors are activated [17]. Limiting variability in the state of the nervous system is essential when assessing the reflexes of those with neuropathology as neuronal pathways are compromised and thus the maintenance of consistent supraspinal input is imperative to accurately eliciting and assessing reflex responses [17]. Further, prolonged testing may also result in outcomes that are confounded by differences in wakefulness [26]. Reducing the time necessary to acquire H-reflex modulation data

would decrease participant and researcher burden, contribute to more comfortable and efficient testing protocols, and improve quality of data. Based on the finding that a 5-pulse train offers similar insight on reflex modulation to the 11-pulse train, we recommend the use of a 5-pulse train opposed to longer multi-pulse trains.

## Limitations

This study offers findings that are specific to people with incomplete SCI and, as a result, a limitation of this study is that the findings may not be generalizable to other neurologic populations. Additionally, the participants in this study had a wide range of injury severities and clinical presentations that may have contributed to inter-participant variability.

The application of these findings is limited to LFD protocols utilizing 1 Hz of stimulation as there are multiple physiological mechanisms operating at different timepoints believed to be contributing to alterations in reflex excitability [24]. Changing the interpulse interval could result in the evaluation of mechanisms where these findings may not apply.

## Conclusions

Assessing spinal reflex excitability in persons with neurologic conditions is increasingly important for identifying optimal neuromodulation approaches. While abbreviated trains do not elicit the same magnitude of reflex depression observed with the 11-pulse LFD train in persons with incomplete SCI, an abbreviated, 5-pulse LFD protocol may offer insight similar to that obtained through longer LFD trains, while improving efficiency and reducing participant burden. We recommend the use of an abbreviated 5-pulse LFD train in lieu of longer, more time-consuming trains.

## Supporting information

**S1 Data.**
(XLSX)

## Acknowledgments

We would like to thank the participants who volunteered their time to participate in the study. We also thank the following who made this study possible: Stephen Estes; Jennifer Iddings; Evan Sandler; Teresa Snow.

## Author Contributions

**Conceptualization:** Jasmine M. Hope, Anastasia Zarkou, Edelle C. Field-Fote.

**Data curation:** Anastasia Zarkou.

**Formal analysis:** Charles J. Creech.

**Funding acquisition:** Edelle C. Field-Fote.

**Investigation:** Jasmine M. Hope, Anastasia Zarkou.

**Methodology:** Charles J. Creech, Edelle C. Field-Fote.

**Supervision:** Jasmine M. Hope, Anastasia Zarkou, Edelle C. Field-Fote.

**Validation:** Charles J. Creech.

**Writing – original draft:** Charles J. Creech, Jasmine M. Hope.

**Writing – review & editing:** Charles J. Creech, Jasmine M. Hope, Anastasia Zarkou, Edelle C. Field-Fote.

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
