## [Decision Letter · Decision Letter 0]

21 Dec 2023

PONE-D-23-32555Optimizing Assessment of Neuromodulation in Persons with Spinal Cord InjuryPLOS ONE

Dear Dr. Creech,

Thank you for submitting your manuscript to PLOS ONE. After careful consideration, we feel that it has merit but does not fully meet PLOS ONE’s publication criteria as it currently stands. Therefore, we invite you to submit a revised version of the manuscript that addresses the points raised during the review process.

We look forward to receiving your revised manuscript.

Kind regards,

Alexander Rabchevsky, Ph.D.

Academic Editor

PLOS ONE

Reviewers' comments:

Reviewer's Responses to Questions

**Comments to the Author**

1. Is the manuscript technically sound, and do the data support the conclusions?

Reviewer #1: Yes

Reviewer #2: Partly

Reviewer #3: Yes

2. Has the statistical analysis been performed appropriately and rigorously? 

Reviewer #1: Yes

Reviewer #2: No

Reviewer #3: Yes

3. Have the authors made all data underlying the findings in their manuscript fully available?

Reviewer #1: Yes

Reviewer #2: Yes

Reviewer #3: No

4. Is the manuscript presented in an intelligible fashion and written in standard English?

Reviewer #1: Yes

Reviewer #2: Yes

Reviewer #3: Yes

5. Review Comments to the Author

Reviewer #1: The current study is on a topic of electrophysiological testing for spasticity in people with spinal cord injury. This topic is very important for collecting meaningful outcomes efficiently in clinical research. This study also highlights the understudied area in the methodological barriers of traditional testing. As a research paper, this manuscript is very well written. There is some critical information missing as I pointed out below. A minor revision by adding methodological details and discussion is recommended for considering publication in this journal.

Review comments:

Title: As the authors alluded throughout the manuscript, this result can be applied to only people with incomplete SCI and 1Hz stimulation. This study was not exactly assessing neuromodulation yet while it can be used in the future after further validation. The current title misleads audience. I recommend that the authors changes the title to “Optimizing Assessment of Low Frequency H-reflex Depression in Persons with Spinal Cord Injury.“

Page 5-6: Add how to determine which leg that received the stimulation and recording. I assume depending on dominant or more spasticity.

Page 7: the following sentence needs to be moved before the Page 6 last paragraph to define H1. “The H-reflex evoked by the first stimulus was considered the control reflex (H1). The 10 reflexes elicited following H1 were considered conditioned reflexes (H2, H3, etc.).”

Page 7: The authors might want to add more details of EMG analysis including bandpass filter, software they used, etc.

Page 8 line 140: The authors need to justify the following statement by references. Friedman test would be recommended based on the sample size.

“Despite this, we utilized the repeated measures ANOVA as it is robust against violations of normality.”

Page 6-7: There is no paired pulse depression study protocol was mentioned. It was implied later in the data analysis, but it has to be added following the LFD protocol.

Discussion

The authors may want to discuss or add limitations of grouping incomplete SCI. There are heterogeneity of spasticity and motoneuron function (peripheral nerve health, muscle health) which likely impact the results in this study participant group.

Paired pulse depression with shorter intervals (i.e., 60ms) can be used for a different physiology such as presynaptic inhibition, so that this 5 pulses rule may not be applicable. The authors need to address this in the discussion.

We do not know whether very high correlation indicates the sensitivity of 5 train LFD is enough to detect a therapeutic efficacy for spasticity in some other treatment such as spinal cord stimulation. The authors need to acknowledge the use of LFD and PPD as outcomes for other therapeutic modalities and discuss the study limitation about very specific 1Hz stimulation study design.

Reviewer #2: The authors present results from a sub-study of 20 individuals with incomplete spinal cord injury to compare H-reflex depression following various abbreviated trains to an 11-pulse train to see if a shorter evaluation could be utilized to study neuromodulation. Authors conclude that a 5-pulse train elicits similar results to the full 11-pulse train and recommend it for use. The manuscript will be strengthened if the authors consider the following points:

1. Authors should provide more detail about the repeated measures ANOVA that were used to assess M-Wave reliability. For example, were separate analyses performed for each individual across the pulses or were all individuals analyzed together? What factors were included?

2. Given the small sample size, Table 2 should include 95% confidence intervals for the Mean difference vs LFD and for the Pearson correlation coefficient. The R-squared column can be removed, since someone can easily compute that if they want.

3. To further provide evidence of the similarity between the 5-pulse train and the 11-pulse LFD, authors should add a scatter plot of the two measures, so that readers can see the distribution/spread of values.

Minor points:

1. line 82 - should "knees fully extended" be "legs fully extended"?

2. line 135: "The data was" should be "The data were"

3. lines 138-139, I'm guessing P40, P65, etc. refer to certain participant IDs, though that is not immediately clear. Authors should clarify that. Since these violate the assumption of normality, it might be worth including the H- and M- wave tracings for one of those participants in supplemental material for comparison with Figure 1.

Reviewer #3: This report examines the number of pulses that are optimal to evoke low-frequency depression in subjects with incomplete spinal cord injury. Whereas typically, 11 pulses are used, the authors suggest fewer pulses are almost as effective, ie. five as compared to 11. In addition, the authors compared two pulses in a paired-pulse paradigm with serial pulses.

It seems interesting to understand the variations between individual individuals and somewhat surprising that there would not be an individually optimized number of pulses.

While this data is quite convincing, it is a fairly simple technical observation, and its overall impacts on the field are not evident other than influencing people to explore shorter pulse trains. This may shorten sessions with research subjects and reduce the burden in H-reflex testing.

It may be helpful to indicate why some individuals have any evocable H- reflex, and others do not after spinal cord injury. For example, there is no apparent difference in the clinical parameters provided.

It's not clear in Figure 2 which variability measured is being shown- standard deviation versus standard error?

6. PLOS authors have the option to publish the peer review history of their article (what does this mean?). If published, this will include your full peer review and any attached files.

Reviewer #1: No

Reviewer #2: No

Reviewer #3: **Yes: **James D Guest

---

## [Author Response · Author response to Decision Letter 0]

2 Feb 2024

We thank the reviewers for their time and appreciate their valuable comments. We have made a number of edits to the manuscript to reflect the input from the reviewers. Below, you will find responses to individual comments. Comments by the authors are included in italics.

Reviewer #1: The current study is on a topic of electrophysiological testing for spasticity in people with spinal cord injury. This topic is very important for collecting meaningful outcomes efficiently in clinical research. This study also highlights the understudied area in the methodological barriers of traditional testing. As a research paper, this manuscript is very well written. There is some critical information missing as I pointed out below. A minor revision by adding methodological details and discussion is recommended for considering publication in this journal.

Review comments:

Title: As the authors alluded throughout the manuscript, this result can be applied to only people with incomplete SCI and 1Hz stimulation. This study was not exactly assessing neuromodulation yet while it can be used in the future after further validation. The current title misleads audience. I recommend that the authors changes the title to “Optimizing Assessment of Low Frequency H-reflex Depression in Persons with Spinal Cord Injury.“

We concede to the reviewer’s request and have changed the title as requested to “Optimizing Assessment of Low Frequency H-Reflex Depression in Persons with Spinal Cord Injury.”

Page 5-6: Add how to determine which leg that received the stimulation and recording. I assume depending on dominant or more spasticity.

The reviewer is correct; we selected the tested leg based on participant self-report of the more spastic leg. This information has been added on page 5, lines 79-80, reading as follows;

“The more spastic leg as self-reported by participants was assessed during testing.”

Page 7: the following sentence needs to be moved before the Page 6 last paragraph to define H1. “The H-reflex evoked by the first stimulus was considered the control reflex (H1). The 10 reflexes elicited following H1 were considered conditioned reflexes (H2, H3, etc.).”

We appreciate this recommendation as the abbreviation “H1” is utilized before it was defined. The statement was moved to the paragraph at the top of page 7 (line 110) to define the abbreviated H-reflex nomenclature (I.e. H1, H2, etc).

Page 7: The authors might want to add more details of EMG analysis including bandpass filter, software they used, etc.

The requested information is included under the subtitle, “Electrophysiologic measurements” on page 5. No filtering was completed after the signal was collected before we measured the peak-to-peak amplitude. We have added the pre-amplification value of x20 (line 84). 

Page 8 line 140: The authors need to justify the following statement by references. Friedman test would be recommended based on the sample size.

“Despite this, we utilized the repeated measures ANOVA as it is robust against violations of normality.”

We concede to the reviewer’s request, changing the reported analysis to the Friedman test. The new information can be found on page 8, line 135. The outcome of the analysis does not change (i.e. no significant differences were observed between M-waves). 

Page 6-7: There is no paired pulse depression study protocol was mentioned. It was implied later in the data analysis, but it has to be added following the LFD protocol.

All data was collected using the methodology described for low-frequency depression, but the calculation methodology varied between assessments. For example, 11 pulse trains were always utilized, but only reflex amplitudes for the 1st and 2nd responses were utilized for the calculation of PPD. The same applies for the 3-11 pulse trains. This information is included on page 7, lines 120-125. We have added the following text early in the “Electrophysiologic measurement” section (lines 108-109):

“The data collected during this electrophysiologic testing was utilized for the calculation of LFD, PPD, and depression of abbreviated trains.”

Discussion

The authors may want to discuss or add limitations of grouping incomplete SCI. There are heterogeneity of spasticity and motoneuron function (peripheral nerve health, muscle health) which likely impact the results in this study participant group.

We agree that there is significant heterogeneity that can occur within our participant group that may influence outcomes, we have addressed this in the limitations section (page 14). 

Paired pulse depression with shorter intervals (i.e., 60ms) can be used for a different physiology such as presynaptic inhibition, so that this 5 pulses rule may not be applicable. The authors need to address this in the discussion.

We have added a statement to the limitations of the study to acknowledge that any physiologic interpretations made with our data are limited to mechanisms that might be affected with a 1-second interpulse interval. The statement can be found on page 14 (lines 244-247) and reads as follows:

“The application of these findings is limited to LFD protocols utilizing 1 Hz of stimulation as there are multiple physiological mechanisms operating at different timepoints believed to be contributing to alterations in reflex excitability (24). Changing the interpulse interval could result in the evaluation of mechanisms where these findings may not apply.”

We do not know whether very high correlation indicates the sensitivity of 5 train LFD is enough to detect a therapeutic efficacy for spasticity in some other treatment such as spinal cord stimulation. The authors need to acknowledge the use of LFD and PPD as outcomes for other therapeutic modalities and discuss the study limitation about very specific 1Hz stimulation study design.

The utilization of low frequency depression of the H-reflex is primarily to assess only some of the mechanisms believed to be responsible for spasticity. Additionally, it is well-accepted that people with spinal cord injury have impaired depression of this reflex when assessed via low frequency depression (Schindler-Ivens S, Shields RK, Exp Brain Res, 2000). Alterations in H-reflex depression, however, may not correlate well with functional assessments of spasticity (Kohan, Acta Med Iran, 2010) so it is unclear how much reflex depression should be observed before therapeutic efficacy can be deduced. Through demonstrating a very strong correlation between the abbreviated 5-pulse train and 11-pulse train LFD, it can be concluded that the 5-pulse train results in data that enables similar conclusions to be made as with 11-pulse LFD, as there is a very strong correlation between the two. 

We agree that there is value to more clarification in regard to the limitation of when this methodology can be applied. Above comments were integrated into the manuscript to clarify this important point- primarily referring to the 1Hz stimulation frequency. Within the manuscript we note that this technique could be used to assess neuromodulation intervention, but refer to transcutaneous spinal stimulation and epidural spinal stimulation as they are most prevalent in current research. In the introduction (page 3, line 25), we have edited the wording to clarify that this assessment technique can be used to assess neuromodulation following any therapeutic modality that may alter monosynaptic reflex excitability.

Reviewer #2: The authors present results from a sub-study of 20 individuals with incomplete spinal cord injury to compare H-reflex depression following various abbreviated trains to an 11-pulse train to see if a shorter evaluation could be utilized to study neuromodulation. Authors conclude that a 5-pulse train elicits similar results to the full 11-pulse train and recommend it for use. The manuscript will be strengthened if the authors consider the following points:

1. Authors should provide more detail about the repeated measures ANOVA that were used to assess M-Wave reliability. For example, were separate analyses performed for each individual across the pulses or were all individuals analyzed together? What factors were included?

Based on feedback from reviewers 1 and 2, we have changed our analysis of M-wave amplitude to a Friedman test. The analysis continues to assess the same variables and the results are the same. All participants were analyzed together at each elicited reflex to ensure that the average M-waves preceding each recorded H-reflex were not significantly different. M-Wave amplitudes recorded prior to each of the elicited reflexes in the LFD train were compared to ensure that similar motor units were being recruited during assessments. 

The referenced statement was edited to read as follows (page 7, lines 134-136):

“To verify that the stimuli were recruiting similar numbers of motor neurons over the course of the conditioning train, a Friedman test was performed to assess the effect of number of elicited reflexes on M-wave amplitude.”

2. Given the small sample size, Table 2 should include 95% confidence intervals for the Mean difference vs LFD and for the Pearson correlation coefficient. The R-squared column can be removed, since someone can easily compute that if they want.

The 95% confidence interval has been added, as recommended.

3. To further provide evidence of the similarity between the 5-pulse train and the 11-pulse LFD, authors should add a scatter plot of the two measures, so that readers can see the distribution/spread of values.

Please see below the scatter plot recommended (image included in uploaded file). On the X-axis, normalized H-reflex amplitude by the traditional 11-pulse train (LFD) is included. On the Y axis, depression elicited by both the 2-pulse train (paired-pulse depression) and 5-pulse train (recommended) are included. We also included a line demonstrating a perfect correlation (1).

It does seem that the distribution/spread of these values decreases with our recommended train, however we believe that the provided correlation coefficients offer sufficient information to draw similar conclusions.

Minor points:

1. line 82 - should "knees fully extended" be "legs fully extended"?

Yes, thank you. Change made as recommended.

2. line 135: "The data was" should be "The data were"

Thank you. The corrected line was removed due to changes in the analysis utilized. 

3. lines 138-139, I'm guessing P40, P65, etc. refer to certain participant IDs, though that is not immediately clear. Authors should clarify that. Since these violate the assumption of normality, it might be worth including the H- and M- wave tracings for one of those participants in supplemental material for comparison with Figure 1.

This is a valid recommendation. We removed the participant abbreviation (PXX) and replaced it with “participant xx” (all participant IDs are included in the demographics table). 

Based on feedback provided by reviewers 1 and 2, we compared our M-waves with a Friedman test. The Friedman test does not require tests of normality to assess appropriateness, so these normality analyses were removed. 

Reviewer #3: This report examines the number of pulses that are optimal to evoke low-frequency depression in subjects with incomplete spinal cord injury. Whereas typically, 11 pulses are used, the authors suggest fewer pulses are almost as effective, ie. five as compared to 11. In addition, the authors compared two pulses in a paired-pulse paradigm with serial pulses.

It seems interesting to understand the variations between individual individuals and somewhat surprising that there would not be an individually optimized number of pulses.

We very much agree that a personalized protocol would be the most accurate way of assessing neuromodulation via low frequency depression. Despite this, identifying the optimal number of pulses would still require investigators to collect 11 reflex responses to identify which abbreviated protocol best correlates for that person. The aim of this study was to identify a protocol that could be generalized to individuals within our population. 

While this data is quite convincing, it is a fairly simple technical observation, and its overall impacts on the field are not evident other than influencing people to explore shorter pulse trains. This may shorten sessions with research subjects and reduce the burden in H-reflex testing.

While the observation is simple, a simple solution that reduces participant burden and improves data collection efficiency is of value to the field. As scientists, it is important that we continue to re-evaluate our methodologies to maximize efficiency without compromising data quality. It is possible that there are individuals in whom data may be compromised due to their individual response to the extended data collection periods of current methodologies. Beyond participant burden and researcher efficiency, a reduction in the time and stimuli requirement could decrease risk of changes in wakefulness and decrease risk of spastic response/muscle tone, resulting in improved data quality. 

It may be helpful to indicate why some individuals have any evocable H- reflex, and others do not after spinal cord injury. For example, there is no apparent difference in the clinical parameters provided.

Within our study sample, we were able to evoke an H-reflex in every participant. An edit to clarify that elicited amplitudes needed to meet specific amplitude criteria was included under Materials and methods (page 5, lines 71-72). Some of our participants had a control H-reflex that fell outside the 10-30% range of the M-max that qualified the data to be analyzed. Many of the participants whose data did not meet our criteria for analysis produced H-reflexes that were close to threshold and inconsistent. While this could be an interesting topic of discussion, it is outside the scope of this paper. 

It's not clear in Figure 2 which variability measured is being shown- standard deviation versus standard error? 

Thank you for this valuable observation. They are standard deviation values. This information has been added to the figure description for improved clarity.

---

## [Editor Report · Decision Letter 1]

21 Feb 2024

Optimizing assessment of low frequency H-reflex depression in persons with spinal cord injury

PONE-D-23-32555R1

Dear Dr. Creech,

We’re pleased to inform you that your manuscript has been judged scientifically suitable for publication and will be formally accepted for publication once it meets all outstanding technical requirements. We thank you for your replies to the reviewers and addition of requested documentation to highlight the importance of your findings.

Kind regards,

Alexander Rabchevsky, Ph.D.

Academic Editor

PLOS ONE
---

## [Editor Report · Acceptance letter]

30 Apr 2024

PONE-D-23-32555R1 

PLOS ONE

Dear Dr. Creech, 

I'm pleased to inform you that your manuscript has been deemed suitable for publication in PLOS ONE. Congratulations! Your manuscript is now being handed over to our production team.

Kind regards, 

on behalf of

Dr. Alexander Rabchevsky 

Academic Editor

PLOS ONE